# Contemporary Symbolic Regression Methods and their Relative Performance

**William La Cava**[*]
Boston Children's Hospital
Harvard Medical School
william.lacava@childrens.harvard.edu

**Patryk Orzechowski** [†]
Institute for Biomedical Informatics
University of Pennsylvania
patryk.orzechowski@gmail.com

**Bogdan Burlacu**
Josef Ressel Center for Symbolic Regression
University of Applied Sciences Upper Austria
bogdan.burlacu@fh-ooe.at

**Fabrício Olivetti de França**
Federal University of ABC[‡]
Santo Andre, Brazil
folivetti@ufabc.edu.br

**Marco Virgolin** [§]
Life Sciences and Health Group
Centrum Wiskunde & Informatica
marco.virgolin@cwi.nl

**Ying Jin**
Department of Statistics
Stanford University
ying531@stanford.edu

**Michael Kommenda**
Josef Ressel Center for Symbolic Regression
University of Applied Sciences Upper Austria
michael.kommenda@fh-ooe.at

**Jason H. Moore**
Institute for Biomedical Informatics
University of Pennsylvania
jhmoore@upenn.edu

## Abstract

Many promising approaches to symbolic regression have been presented in recent years, yet progress in the field continues to suffer from a lack of uniform, robust, and transparent benchmarking standards. We address this shortcoming by introducing an open-source, reproducible benchmarking platform for symbolic regression. We assess 14 symbolic regression methods and 7 machine learning methods on a set of 252 diverse regression problems. Our assessment includes both real-world datasets with no known model form as well as ground-truth benchmark problems. For the real-world datasets, we benchmark the ability of each method to learn models with low error and low complexity relative to state-of-the-art machine learning methods. For the synthetic problems, we assess each method's ability to find exact solutions in the presence of varying levels of noise. Under these controlled experiments, we conclude that the best performing methods for real-world regression combine genetic algorithms with parameter estimation and/or semantic search drivers. When tasked with recovering exact equations in the presence of noise, we find that several approaches perform similarly. We provide a detailed guide to reproducing this experiment and contributing new methods, and encourage other researchers to collaborate with us on a common and living symbolic regression benchmark.

[*]corresponding author. Formerly Institute for Biomedical Informatics, University of Pennsylvania

[†]Department of Automatics and Robotics, AGH University of Science and Technology, Krakow, Poland

[‡]Center for Mathematics, Computation and Cognition | Heuristics, Analysis and Learning Laboratory

[§]Formerly (during preparation of this paper) at Chalmers University of Technology, Sweden

35th Conference on Neural Information Processing Systems (NeurIPS 2021) Track on Datasets and Benchmarks.

# 1 Introduction

Symbolic regression (SR) is an approach to machine learning (ML) in which both the parameters and structure of an analytical model are optimized. SR can be useful when one wishes to describe a process via a mathematical expression, especially a simple expression; thus, it is often applied in the hopes of producing a model of a process that, by virtue of its simplicity, may be easy to interpret. Interpretable ML is becoming increasingly important as model deployments in high stakes societal applications such as finance and medicine grow [1, 2]. Moreover, the mathematical expressions produced by SR are well-suited to be analyzed and controlled for their out-of-distribution behavior (e.g., in terms of asymptotic behavior, periodicity, etc.). These attractive properties of SR have led to its application in a number of areas, such as physics [3], biology [4], clinical informatics [5], climate modeling [6], finance [7], and many fields of engineering [8–10].

SR literature has, in general, fallen short of evaluating and ranking new methods in a way that facilitates their widespread adoption. Our view is that this shortcoming largely stems from a lack of standardized, transparent and reproducible benchmarks, especially those that test a large and diverse array of problems [11]. Although community surveys [11, 12] have led to suggestions for improving benchmarking standards, and even black-listed certain problems, contemporary literature continues to be published that violates those standards. Absent these standards, it is difficult to assess which methods or family of methods should be considered "state-of-the-art" (SotA).

Achieving a fleeting sense of SotA is certainly not the singular pursuit of methods research, yet without common, robust benchmarking studies, promising avenues of investigation cannot be well-informed by empirical evidence. We hope the benchmarking platform introduced in this paper improves the cross-pollination between research communities interested in SR, which include evolutionary computation, physics, engineering, statistics, and more traditional machine learning disciplines.

In this paper, we describe a large benchmarking effort that includes a dataset repository curated for SR, as well as a benchmarking library designed to allow researchers to easily contribute methods. To achieve this, we incorporated 130 datasets with ground truth forms into the Penn Machine Learning Benchmark (PMLB) [13], including metadata describing the underlying equations, their units, and various summary statistics. Furthermore, we created a SR benchmark repository called SRBench[5] and sought contributions from researchers in this area. Here we describe this process and the results, which consist of comparisons of 14 contemporary SR methods on hundreds of regression problems.

To our knowledge, this is by far the largest and most comprehensive SR benchmark effort to date, which allows us to make claims concerning current SotA methods for SR with better certainty. Importantly, and in contrast to many previous efforts, the datasets, methods, benchmarking code, and results are completely open-source, reproducible, and revision-controlled, which should allow SRBench to exist as a living benchmark for future studies.

# 2 Background and Motivation

The goal of SR is to learn a mapping $\hat{y}(\mathbf{x}) = \hat{\phi}(\mathbf{x}, \hat{\theta}) : \mathbb{R}^d \to \mathbb{R}$ using a dataset of paired examples $\mathcal{D} = \{(\mathbf{x}_i, y_i)\}_{i=1}^N$, with features $\mathbf{x} \in \mathbb{R}^d$ and target $y$. SR assumes the existence of an analytical model of the form $y(\mathbf{x}) = \phi^*(\mathbf{x}, \theta^*) + \epsilon$ that would generate the observations in $\mathcal{D}$, and seeks to estimate this model by searching the space of expressions, $\phi$, and parameters, $\theta$, in the presence of white noise, $\epsilon$.

---

[5]https://github.com/cavalab/srbench

Koza [14] introduced SR as an application of *genetic programming* (GP), a field that investigates the use of genetic algorithms (GAs) to evolve executable data structures, i.e. programs. In the case of so-called "Koza-style" GP, the programs to be optimized are syntax trees consisting of functions/operations over input features and constants. Like in other GAs, GP is a process that evolves a population of candidate solutions (e.g., syntax trees) by iteratively producing offspring from parent solutions (e.g., by swapping parents' subtrees) and eliminating unfit solutions (e.g., programs with sub-par behavior). Most SR research to date has emerged from within this sub-field and its associated conferences.[6]

Despite the availability of post-hoc methods for explaining black-box model predictions [15], there have been recent calls to focus on learning interpretable/transparent models explicitly [2]. Perhaps due to this renewed interest in model interpretability, entirely different methods for tackling SR have been proposed [16–22]. These include methods based in Bayesian optimization [16], recurrent neural networks (RNNs) [17], and physics-inspired divide-and-conquer strategies [18, 23]. Some of these papers refer to Eureqa, a commercial, GP-based SR method used to re-discover known physics equations [3], as the "gold standard" for SR [17] and/or the best method for SR "by far" [18]. However, Schmidt and Lipson [24] make no claim to being the SotA method for SR, nor is this hypothesis tested in the body of work on which Eureqa is based [25].

Although commercial platforms like Eureqa and Wolfram [26] are successful tools for applying SR, they are not designed to support controlled benchmark experiments, and therefore experiments utilizing them have serious caveats. Due to the design of the front-end API for both tools, it is not possible to benchmark either method against others while holding important parameters of such an experiment constant, including the computational effort, number of model evaluations, CPU/memory limits, and final solution assessment. More generally, researchers cannot uniquely determine which features of the software and/or experiment lead to observed differences in performance, given that these commercial tools are closed-source. In this light, it is not clear what insights are to be gained when comparing to Eureqa and Wolfram beyond a simple head-to-head comparison. Therefore, rather than benchmark against Eureqa in this paper, we implement its underlying algorithms in an open-source package, which allows our experiment to remain transparent, reproducible, accessible, and controlled. We discuss the algorithms underlying Eureqa in detail in Sec. A.3.

A close reading of SR literature since 2009 implies that a number of proposed methods would outperform Eureqa in controlled tests, and are therefore suitable choices for benchmarking (e.g. [27, 28]). Unfortunately, the widespread adoption of these promising SR approaches is hamstrung by a lack of consensus on good benchmark problems, testing frameworks, and experimental designs. Our effort to establish a common benchmark is motivated by our view that common, robust, standardized benchmarks for SR could speed progress in the field by providing a clear baseline from which to assert the quality of new approaches. Consider the NN community's focus on common benchmarks (e.g. ImageNet [29]), frameworks (e.g. TensorFlow, PyTorch) and experiment designs. By contrast, it is common to observe results in SR literature that are based on a small number of low dimensional, easy and unrealistic problems, comparing only to very basic GP systems such as those described in [14] nearly thirty years ago. Despite detailed descriptions of these issues [11], community surveys and proposals to "black-list" toy problems [12], toy datasets and comparisons to out-dated SR methods continue to appear in contemporary literature.

The aspects of performance assessment for SR differ from typical regression benchmarking due to the interest in obtaining concise, symbolic expressions. In general, the trade-off between accuracy and

---

[6]A non-exhaustive list: GECCO, EuroGP, FOGA, PPSN, and IEEE CEC.

simplicity must be considered when evaluating the merits of different models. Furthermore, model *simplicity*, typically measured as sparsity or model size, is but a proxy for model *interpretability*; a simple model may still be un-interpretable, or simply wrong [30–32]. With these concerns in mind, datasets with ground truth solutions are useful, in that they allow researchers to assess whether or not the symbolic model regressed by a given method corresponds to a known analytical solution. Nevertheless, benchmarks utilizing synthetic datasets with ground-truth solutions are not sufficient for assessing real-world performance, and so we consider it essential to also evaluate the performance of SR on real-world or otherwise black-box regression problems, relative to SotA ML methods.

There have been a few recent efforts to benchmark SR algorithms [33], including a precursor to this work benchmarking four SR methods on 94 regression problems [34]. In both cases, SR methods were assessed solely on their ability to make accurate predictions. In contrast, Udrescu and Tegmark [18] proposed 120 new synthetic, physics-based datasets for SR, but compared only to Eureqa and only in terms of solution rates. A major contribution of our work is its significantly more comprehensive scope than previous studies. We include 14 SR methods on 252 datasets in comparison to 7 ML methods. Our metrics of comparison are also more comprehensive, and include 1) accuracy, 2) simplicity, and 3) exact or approximate symbolic matches to the ground truth process. Furthermore, we have made the benchmark openly available, reproducible, and open for contributions supported by continuous integration [35].

## 3   SRBench

We created SRBench to be a reproducible, open-source benchmarking project by pulling together a large set of diverse benchmark datasets, contemporary SR methods, and ML methods around a shared model evaluation and analysis environment. SRBench overcomes several of the issues in current benchmarking literature as described in Sec. 2. For example, it makes it easy for methodologists to benchmark new algorithms over hundreds of problems, in comparison to strong, contemporary reference methods. These improvements allow us to reason with more certainty than in previous work about the SotA methods for SR.

In order to establish common datasets, we extended PMLB, a repository of standardized regression and classification problems [13, 36], by adding 130 SR datasets with known model forms. PMLB provides utilities for fetching and handling data, recording and visualizing dataset metadata, and contributing new datasets. The SR methods we benchmarked are all contemporary implementations (2011 - 2020) from several method families, as shown in Table 1. We required contributors to implement a minimal, Scikit-learn compatible [37], Python API for their method. In addition, contributors were required to provide the final fitted model as a string that was compatible with the symbolic mathematics library sympy. Note that although we require a Python wrapper, SR implementations in many different languages are supported, as long as the Python API is available and the language environment can be managed via Anaconda[7].

To ensure reproducibility, we defined a common environment (via Anaconda) with fixed versions of packages and their dependencies. In contrast to most SR studies, the full installation code, experiment code, results and analysis are available via the repository for use in future studies. To make SRBench as extensible as possible, we automated the process of incorporating new methods and results into the analysis pipeline. The repository accepts rolling contributions of new methods that meet the minimal API requirements. To achieve this, we created a continuous integration (CI) [35] framework that assures contributions are compatible with the benchmark code as they arrive. CI also supports

---

[7]https://www.anaconda.com/

Table 1: Short descriptions of the SR methods benchmarked in our experiment, including references and links to implementations.

| Method | Year | Description | Method Family | Implementation |
|---|---|---|---|---|
| AFP [38] | 2011 | Age-fitness Pareto Optimization | GP | C++/Python (link) |
| AFP_FE [24] | 2011 | AFP with co-evolved fitness estimates; Eureqa-esque | GP | C++/Python (link) |
| AIFeynman [23] | 2020 | Physics-inspired method | Divide and conquer | Fortran/Python (link) |
| BSR [16] | 2020 | Bayesian Symbolic Regression | Markov Chain Monte Carlo | Python (link) |
| DSR [17] | 2020 | Deep Symbolic Regression | Recurrent neural networks | Python (PyTorch) (link) |
| EPLEX [39] | 2016 | $\epsilon$-lexicase selection | GP | C++/Python (link) |
| FEAT [40] | 2019 | Feature Engineering Automation Tool | GP | C++/Python (link) |
| FFX [41] | 2011 | Fast function extraction | Random search | C++/Python (link) |
| GP-GOMEA [42] | 2020 | GP version of the Gene-pool Optimal Mixing Evolutionary Algorithm | GP | C++/Python (link) |
| gplearn | 2015 | Koza-style symbolic regression in Python | GP | C++/Python (link) |
| ITEA [43] | 2020 | Interaction-Transformation EA | GP | Haskell/Python (link) |
| MRGP [44] | 2014 | Multiple Regression Genetic Programming | GP | Java (link) |
| Operon [45] | 2019 | SR with Non-linear least squares | GP | C++/Python (link) |
| SBP-GP [46] | 2019 | Semantic Back-propagation Genetic Programming | GP | C++/Python (link) |

Table 2: Settings used in the benchmark experiments. "Total comparisons" refers to the total evaluatons of an algorithm on a dataset for a given noise level and random seed.

| Setting | Black-box Problems | Ground-truth Problems |
|---|---|---|
| No. of datasets | 122 | 130 |
| No. of algorithms | 21 (14 SR, 7 ML) | 14 |
| No. of trials per dataset | 10 | 10 |
| Train/test split | .75/.25 | .75/.25 |
| Hyperparameter tuning | 5-fold Halving Grid Search CV | Tuned set from black-box problems |
| Termination criteria | 500k evaluations/train or 48 hours | 1M evaluations or 8 hours |
| Levels of target noise | None | 0, 0.001, 0.01, 0.1 |
| Total comparisons | 26840 | 54600 |
| Computing budget | 1.29M core hours | 436.8K core hours |

continuous updates to results reporting and visualization whenever new experiments are available, allowing us to maintain a standing leader-board of contemporary SR methods. Ideally these features will quicken the adoption of SotA approaches throughout the SR research community. Further details on how to use and contribute to SRBench are provided in Sec. A.1.

## 4 Experiment Design

We evaluated SR methods on two separate tasks. First, we assessed their ability to make accurate predictions on "black-box" regression problems (in which the underlying data generating function remains unknown) while minimizing the complexity of the discovered models. Second, we tested the ability of each method to find exact solutions to synthetic datasets with known, ground-truth functions, originating from physics and various fields of engineering.

The basic experiment settings are summarized in Table 2. Each algorithm was trained on each dataset (and level of noise, for ground-truth problems) in 10 repeated trials with a different random state that controlled both the train/test split and the seed of the algorithm. Datasets were split 75/25% in training and testing. For black-box regression problems, each algorithm was tuned using 5-fold cross validation with halving grid search. The SR algorithms were limited to 6 hyperparameter combinations; the ML methods were allowed more, as shown in Table 4-6. The best hyperparameter settings were used to tune a final estimator and evaluate it according to the metrics described above. Details for running the experiment are given in Sec. A.1.

## 4.1 Symbolic Regression Methods

Here we characterize the SR methods summarized in Table 1 by describing how they fit into broader research trends within the SR field. The most traditional implementation of GP-based SR we test is **gplearn**, which initializes a random population of programs/models, and then iterates through the steps of tournament selection, mutation and crossover.

Pareto optimization methods [8, 47–49] are popular evolutionary strategies that exploit Pareto dominance relations to drive the population of models towards a set of efficient trade-offs between competing objectives. Half of the SR methods we test use Pareto optimization in some form during training. Age-Fitness Pareto optimization (**AFP**), proposed by Eureqa's authors Schmidt and Lipson [38], uses a model's age as an objective in order to reduce premature convergence as well as bloat [50]. **AFP_FE** combines AFP with Eureqa's method for fitness estimation [51]. Thus we expect AFP_FE and AFP to perform similarly to Eureqa as described in literature.

Another promising line of research has been to leverage program *semantics* (in this case, the equation's intermediate and final outputs over training samples) more heavily during optimization, rather than compressing that information into aggregate fitness values [52]. $\epsilon$-lexicase selection (**EPLEX**) [27] is a parent selection method that utilizes semantics to conduct selection by filtering models through randomized subsets of cases, which rewards models that perform well on difficult regions of the training data. EPLEX is also used as the parent selection method in FEAT [40]. Semantic backpropagation (SBP) is another semantic technique to compute, for a given target value and a tree node position, that value which makes the output of the model match the target (i.e., the label) [53–55]. Here, we evaluate the (**SBP-GP**) algorithm by Virgolin et al. [46] which improves SBP-based recombination by dynamically adapting intermediate outputs using affine transformations.

Backpropagation-based gradient descent was proposed for GP-SR by Topchy and Punch [56], but tends to appear less often than stochastic hill climbing (e.g. [3, 57]). More recent studies [45, 58] have made a strong case for the use of gradient-based constant optimization as an improvement over stochastic and evolutionary approaches. The aforementioned studies are embodied by **Operon**, a GP method that incorporates non-linear least squares constant optimization using the Levenberg-Marquadt algorithm [59].

In addition to the question of how to best optimize constants, a line of research has proposed different ways of defining program and/or model encodings. The methods FEAT, MRGP, ITEA, and FFX each impose additional structural assumptions on the models being evolved. In **FEAT**, each model is a linear combination of a set of evolved features, the parameters of which are encoded as edges and optimized via gradient descent. In **MRGP** [44], the entire program trace (i.e., each subfunction of the model) is decomposed into features and used to train a Lasso model. In **ITEA**, each model is an affine combination of *interaction-transformation* expressions, which compose a unary function (the transformation) and a polynomial function (the interaction) [43, 60]. Finally, **FFX** [41] simply initializes a population of equations, selects the Pareto optimal set, and returns a single linear model by treating the population of equations as features.

**GP-GOMEA** is a GP algorithm where recombination is adapted over time [42, 61]. Every generation, GP-GOMEA builds a statistical model of interdependencies within the encoding of the evolving programs, and then uses this information to recombine interdependent blocks of components, as to preserve their concerted action.

Jin et al. [16] recently proposed Bayesian Symbolic Regression (**BSR**), in which a prior is placed on tree structures and the posterior distributions are sampled using a Markov Chain Monte Carlo

(MCMC) method. As in GP-based SR, arithmetic expressions are expressed with symbolic trees, although BSR explicitly defines the final model form as a linear combination of several symbolic trees. Model parsimony is encouraged by specifying a prior that presumes additive, linear combinations of small components.

Deep Symbolic Regression (**DSR**) [17] uses reinforcement learning to train a generative RNN model of symbolic expressions. Expressions sampled from the model distribution are assessed to create a reward signal. DSR introduces a variant of the Monte Carlo policy gradient algorithm [62] dubbed a "risk-seeking policy gradient" in an effort to bias the generative model towards exact expressions.

**AIFeynman** is a divide-and-conquer approach that recursively applies a set of solvers and problem decomposition heuristics to build a symbolic model [18]. If the problem is not directly solve-able by polynomial fitting or brute-force search, AIFeynman trains a NN on the data and uses it to estimate functional modularities (e.g., symmetry and/or separability), which are used to partition the data into simpler problems and recurse. An updated version of the algorithm, which we test here, integrates Pareto optimization with an information-theoretic complexity metric to improve robustness to noise [23].

## 4.2 Datasets

All of the benchmark datasets are summarized by number of instances and number of features in Fig. 5. The problems range from 47 to 1 million instances, and two to 124 features. We used 122 black-box regression problems available in PMLB v.1.0. These problems are pulled from, and overlap with, various open-source repositories, including OpenML [63] and the UCI repository [64]. PMLB standardizes these datasets to a common format and provides fetching functions to load them into Python (and R). The black-box regression datasets consist of 46 "real-world" problems (i.e., observational data collected from physical processes) and 76 synthetic problems (i.e., data generated computationally from static functions or simulations). The black-box problems cover diverse domains, including health informatics (11), business (10), technology (10), environmental science (11) and government (12); in addition, they are derived from varied data sources, including human subjects (14), environmental observations (11), government studies (12), and economic markets (7). The datasets can be browsed by their properties at epistasislab.github.io/pmlb. Each dataset includes metadata describing source information as well as a detailed profile page summarizing the data distributions and interactions (here is an example).

We extended PMLB with 130 datasets with known, ground-truth model forms. These datasets were used to assess the ability of SR methods to recover known process physics. The 130 datasets came from two sources: the Feynman Symbolic Regression Database, and the ODE-Strogatz repository. Both sets of data come from first principles models of physical systems. The Feynman problems originate in the *Feynman Lectures on Physics* [65], and the datasets were recently created and proposed as SR benchmarks [18]. Whereas the Feynman datasets represent static systems, the Strogatz problems are non-linear and chaotic dynamical processes [66]. Each dataset is one state of a 2-state system of first-order, ordinary differential equations (ODEs). They were used to benchmark SR methods in previous work [25, 67], and are described in more detail in Sec. A.4 and Table 3.

### 4.3 Metrics

**Accuracy**  We assessed accuracy using the coefficient of determination, defined as

$$R^2 = 1 - \frac{\sum_i^N (y_i - \hat{y}_i)^2}{\sum_i^N (y_i - \bar{y}_i)^2}.$$

**Complexity**  A number of different complexity measures have been proposed for SR, including those based on *syntactic* complexity (i.e. related to the complexity of the symbolic model); those based on *semantic* complexity (i.e. related to the behavior of the model over the data) [23, 68]; those using both definitions [69]; and those estimating complexity via meta-learning [70]. The pros and cons of these methods and their relation to notions of interpretability is a point of discussion [71]. For the sake of simplicity, we opted to define complexity as the number of mathematical operators, features and constants in the model, where the mathematical operators are in the set $\{+, -, *, /, \sin, \cos, \arcsin, \arccos, \exp, \log, \mathrm{pow}, \max, \min\}$. In addition to calculating the complexity of the raw model forms returned by each method, we calculated the complexity of the models after simplifying via sympy.

**Solution Criteria**  For the ground-truth regression problems, we used the following solution definition.

**Definition 4.1** (Symbolic Solution). A model $\hat{\phi}(\mathbf{x}, \hat{\theta})$ is a Symbolic Solution to a problem with ground-truth model $y = \phi^*(\mathbf{x}, \theta^*) + \epsilon$, if $\hat{\phi}$ does not reduce to a constant, and if either of the following conditions are true: 1) $\phi^* - \hat{\phi} = a$; or 2) $\phi^*/\hat{\phi} = b, b \neq 0$, for some constants $a$ and $b$.

This definition is designed to capture models that differ from the true model by a constant or scalar. Prior to assessing symbolic solutions, each model underwent sympy simplification, as did the conditions above. Relative to accuracy metrics, the Symbolic Solution metric is a more faithful evaluation of the ability of an SR method to discover the data generating process. However, because models can be represented in myriad ways, and sympy's simplification procedure is non-optimal, we cannot guarantee that all symbolic solutions are captured with perfect fidelity by this metric.

## 5   Results

The median test set performance on all problems and methods for the black-box benchmark problems is summarized in Fig. 1. Across the problems, we find that the models generated by Operon are significantly more accurate than any other method's models in terms of test set $R^2$ ($p \leq$6.5e-05). SBP-GP and FEAT rank second and third and attain similar accuracies, although the models produced by FEAT are significantly smaller ($p =$9.2e-22).

We note that four of the top five methods (Operon, SBP-GP, FEAT, EPLEX) and six of the top ten methods (GP-GOMEA, ITEA) are GP-based SR methods. The other top methods are ensemble tree-based methods, including two popular gradient-boosting algorithms, XGBoost and LightGBM [72, 73]); Random Forest [74]; and AdaBoost [75]. Among these methods, Operon, FEAT and SBP-GP significantly outperform and LightGBM ($p \leq$1.3e-07) and Operon and SBP-GP outperform XGBoost ($p \leq$1.3e-04). We also note ITEA's overall accuracy is not significantly different from RandomForest or AdaBoost. Of note, the models produced by the top five SR methods (aside from SBP-GP) are 1-3 orders of magnitude smaller than models produced by the ensemble tree-based approaches ($p \leq$1.3e-21).

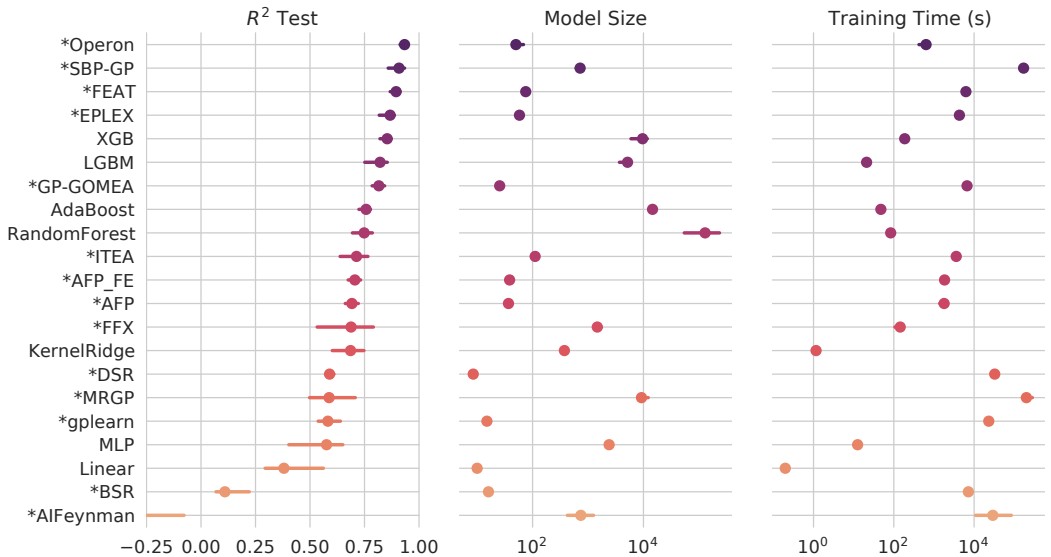

Figure 1: Results on the black-box regression problems. Points indicate the mean of the median test set performance on all problems, and bars show the 95% confidence interval. "*": SR methods.

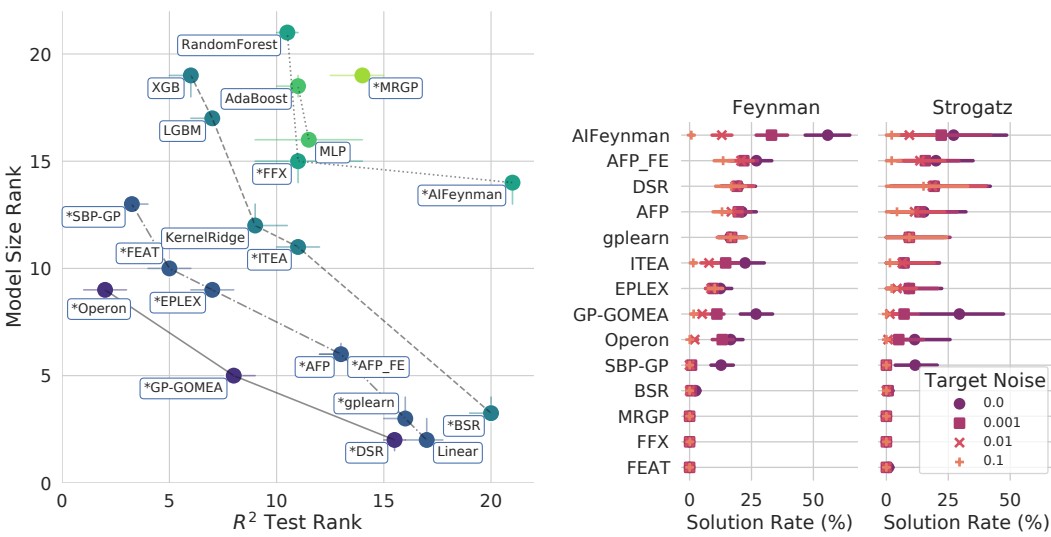

Figure 2: Pareto plot comparing the rankings of all methods in terms of model size and $R^2$ score on the black-box problems. Points denote median rankings and the bars denote 95% confidence intervals. Connecting lines and color denote Pareto dominance rankings. "*": SR methods.

Figure 3: Solution rates for the ground-truth regression problems. Color/shape indicates level of noise added to the target variable.

Among the non-GP-based SR algorithms, FFX and DSR perform similarly to each other ($p =0.76$) and significantly better than BSR and AIFeynman ($p \leq$6.1e-05). FFX trains more quickly than DSR, although DSR produces some of the smallest solutions, akin to penalized regression. We note that AIFeynman performs poorly on these problems, suggesting that not many of them exhibit the qualities of physical systems (rotational/translational invariance, symmetry, etc.) that AIFeynman was designed to exploit. Additional statistical comparisons are given in Figs. 9-11.

In Fig. 2, we illustrate the performance of the methods on the black-box problems when accuracy and simplicity are considered simultaneously. The Pareto front for these two objectives (solid line) is composed of three methods: Operon, GP-GOMEA, and DSR, which taken together give the set of best trade-offs between accuracy and simplicity across the black-box regression problems.

Performance on the ground-truth regression problems is summarized in Fig. 3, with methods sorted by their median solution rate and grouped by data source (Feynman or Strogatz). On average, when the target is free of noise, we observe that AIFeynman identifies exact solutions 53% of the time, nearly twice as often as the next closest method (GP-GOMEA, 27%). However, at noise levels of 0.01 and above, four other methods recover exact solutions more often: DSR, gplearn, AFP_FE, and AFP. Taken together, the black-box and ground-truth regression results suggest AIFeynman may be brittle in application to real-world and/or noisy data, yet its performance with little to no noise is significant for the Feynman problems. On the Strogatz datasets, AIFeynman's performance is not significantly different than other methods, and indeed there are few significant differences in performance between the top 10 methods at any noise level. We note that the top-ranked method on real-world data, Operon, struggles to recover solutions to these problems, despite finding many candidate solutions with near prefect test set scores. See Sec. A.6-A.7 for additional analysis.

## 6    Discussion and Conclusions

This paper introduces a SR benchmarking framework that allows objective comparisons of contemporary SR methods on a wide range of diverse regression problems. We have found that, on real-world and black-box regression tasks, contemporary GP-based SR methods (e.g. Operon) outperform new SR methods based in other fields of optimization, and can also perform as well as or better than gradient boosted trees while producing simpler models. On synthetic ground-truth physics and dynamical systems problems, we have verified that AIFeynman finds exact solutions significantly better than other methods when noise is minimal; otherwise, both deep learning-based methods (DSR) and GP-based SR methods (e.g. AFP_FE) perform best.

We see clear ways to improve SRBench by improving the dataset curation, experiment design and analysis. For one, we have not benchmarked the methods in a setting that allows them to exploit parallelism, which may change relative run-times. There are also many promising SR methods not included in this study that we hope to add in future revisions. In addition, whereas our benchmark includes real-world data as well as simulated data with ground-truth models, it does not include real-world data from phenomena with known, first principles models (e.g., observations of a mass-spring-damper system). Data such as these could help us better evaluate the ability of SR methods to discover relations under real-world conditions. We intend to include these data in future versions, given the evidence that SR models can sometimes discover unexpected analytical models that outperform the expert models in a field (e.g., in studies of yeast metabolism [76] and fluid tank systems [67]). As a final note, our current study highlights orthogonal approaches to SR that show promise, and in future work we hope to explore whether combinations of proposed methods (e.g., non-linear parameter optimization plus semantic search drivers) would have synergistic effects.

## Acknowledgments

William La Cava was supported by the National Library of Medicine and National Institutes of Health under awards K99LM012926 and R00LM012926. He would like to thank Curt Calafut, members of the Epistasis Lab, and Joseph D. Romano for coming through in a pinch.

Ying Jin would like to thank Doctor Jian Guo for hosting an internship for the project and Professor Jian Kang for helpful and inspiring guidance in Bayesian statistics.

Fabricio Olivetti de França was supported by Fundação de Amparo à Pesquisa do Estado de São Paulo (FAPESP), grant number 2018/14173-8.

Patryk Orzechowski and Jason H. Moore were supported by NIH grant LM010098.

The authors would also like to thank contributors to the SRBench repository, including James McDermott and Aurélie Boisbunon. They additionally thank Randal Olson and Weixuan Fu for their initial push to integrate regression benchmarking into PMLB.

This computational experiments were supported by the Penn Medicine Academic Computing Services (PMACS) as well as the PLGrid Infrastructure. Authors declare no competing interests.

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
