# OpenReview forum: "Contemporary Symbolic Regression Methods and their Relative Performance"
_NeurIPS.cc/2021/Track/Datasets_and_Benchmarks/Round1 — NeurIPS 2021 Datasets and Benchmarks Track (Round 1)_

### Official Review · Reviewer_m58C · 2021-07-01
**Straightforward paper introducing benchmark and related experimental results for symbolic regression**

**Rating:** 8
**Confidence:** 2

**Strengths:**

First, a disclaimer: I am not an expert in symbolic regression or genetic programming, so this review is written from the perspective  of a technically literate audience that is interested in but not well-versed in the details of the symbolic regression algorithms presented here. As such, I cannot vouch for the technical accuracy of the descriptions of the algorithms in Section 3.

That said, I appreciate the organization of Section 3. The descriptions clearly reference back to Table 1, which includes both sources for each method and links to its code on GitHub. Clicking, for example, on the “Operon” row link yields a well-formatted Github page with building and usage instructions and example code. These would be valuable resources to the expert or to a person looking to get into this area.

More broadly, the paper and SRBench represent a resource for the SR community. Although the benchmark draws on existing datasets and approaches,  the project of gathering those together into an accessible, reproducible, standardized framework;  incorporating 14 SR methods; and making the whole thing open to future contributions, represents extensive, valuable work. The experimental results may also be useful, both in guiding practitioners in their algorithm choices and as results that other researchers may seek to replicate or extend.

**Weaknesses:**


The introduction and conclusion note that the benchmark provides a “large and diverse” set of regression problems and the paper repeatedly mentions the need to assess “real-world performance,” but it does not make it clear what kind of variety the datasets cover or what ‘real’ means here. The datasets derived from physics equations are clearly more ‘real’ than contrived toy examples, but are they a good representation of the main ‘real’ context that SR is applied to? The authors note in the conclusion that future improvements might incorporate “more realistic applications settings” but the paper would be improved if it gave a bit more attention to this point earlier on.

In particular, the introduction gives little sense of what is at stake in a broader sense in the use of SR methods. I would recommend including some mentions here of the kinds of areas where SR methods have been or could be applied (or perhaps list a few example tasks, more specific than “physics, engineering, statistics…”). In what contexts is it advantageous or necessary to learn an analytical model rather than only predict? Or, do SR algorithms sometimes do better at prediction than ML algorithms and are they being adopted in ML applications? It would also be helpful to have a summary of what the datasets cover. Clearly, the ground-truth datasets come from physics, but what kinds of application areas are the PMLB datasets drawn from? Does it have any relevant gaps, known limitations, or conscious omissions? This can all be fairly brief, but it would help both motivate the paper and make the limitations and social implications of the benchmark more specific.

There is additional detail on the data in the appendix, but it remains unclear to me (A) whether SR is actively being applied, say, in criminal justice applications (the example dataset context mentioned in the appendix) and (B) whether any information warning of potential biases or contextual factors for particular datasets is provided with the datasets.


**Additional Feedback:**

There are two comparisons happening in the paper – an internal one among SR algorithms (some of which have roots in ML methods) and an external one to non-SR ML algorithms. This contrast might be made more explicit. In any case, I would be interested to read more about how SR and ML approaches are similar and different and what SR algorithms add, even on tasks that ML methods perform well on. I leave this here because I do not think adding this to this paper is essential.

----------

*** Tidbits that did not affect my overall review of the paper: ***

This sentence does not make sense to me (pg. 18): “In that regard, we do not foresee it as creating additional ethical issues  around their use, unless our work inspires a significant number of people to research symbolic regression.”

There is a typo on page 6. I believe where it reads “problems (i.e. such, in which the underlying…),” it should simply read “(i.e. in which the…)”


**Clarity:**

The paper is clearly written, with good organization and logical flow.

Figure 1 – reads clearly.  It is easy to evaluate and compare algorithms on multiple criteria at once.

Figure 2- this plot is a little less clear on its own, though the explanation in the text helps.  It is not clear to me whether the shapes and colors of the points have any significance.

Figure 3 – reads clearly.


**Correctness:**

I note again that I am not qualified to judge the correctness of the descriptions of the algorithms in section 3. The datasets are drawn from existing resources, but I have little further sense (from the paper at least) of how they were constructed. The design of the comparison between methods and presentation of the output appear reasonable, and Table 2 (+ the appendix) helpfully summarizes some important settings used. I have no reason to doubt that the resulting comparison of algorithm performance given those settings is inaccurate.

**Documentation:**

The benchmark and the algorithms used appear well-documented on GitHub, with attention to managing packages and dependencies with Anaconda and requiring contributor submissions to be compatible with existing code. Various URLs are embedded in the paper, providing direct access. In addition to GitHub documentation, the appendix provides detail on how to run and contribute to the benchmark as well as detailed information about hyperparameter choices.

**Ethics:**

I am not aware of any pressing issues, though as I noted, it isn't clear to me how much information is provided about biases or contextual factors for the datasets in the existing PMLB benchmark they draw on.

**Relation To Prior Work:**

Yes, the authors compare their work to existing commercial platforms, noting that their benchmark provides better ability to control experiment parameters across methods and thus determine what is really driving performance differences. They note other recent benchmarking work but distinguish their work by its attention to both solution rates and prediction accuracy rather than one or the other. The authors also motivate the paper with references to a 2012 discussion of the need for better benchmarks in genetic programming and refer in a general way to existing bad practices, which it claims have continued since 2012. They do not name particular examples here, but this may be appropriate as it avoids scapegoating particular papers or authors.

**Summary And Contributions:**

The paper is motivated by the need for an accessible, reproducible, and realistic benchmark for symbolic regression (SR) research to help the field evaluate new methods, agree on the quality of existing ones, and reduce reliance on toy problems. To that end, it contributes SRBench, a repository of datasets and 14 SR methods constructed to allow integration of future work. The paper states that this represents “the largest and most comprehensive SR benchmark effort to date.” Relative to previous work, the paper also contributes particular attention to the multiple criteria at play in evaluating SR methods. Because SR involves not just prediction but learning an generative form, these criteria include prediction accuracy, model simplicity, and whether the method learns the true underlying model. For this reason, SRBench augments the existing Penn Machine Learning Benchmark (PMLB) of datasets without specified ground truth with a second set of 130 synthetic datasets from physics that have known ground truth.  The paper also uses the benchmark to compare the performance of the 14 SR methods. It describes statistically significant differences among the algorithms in terms of both prediction and ground-truth, though some algorithms do well for one task and not the other. For the black-boxed prediction tasks, it also compares SR methods to non-SR machine learning methods and finds some SR methods can produce models that are both better and simpler.

---

> ### Author Response · Authors · 2021-07-12
> **motivations for SR and dataset descriptions**
>
> > The introduction and conclusion note that the benchmark provides a “large and diverse” set of regression problems and the paper repeatedly mentions the need to assess “real-world performance,” but it does not make it clear what kind of variety the datasets cover or what ‘real’ means here.
>
> We have added additional context for the black-box regression datasets (lines 225-238) and have clarified what is meant by "real-world" and "synthetic" in the paper (lines 230-232). In addition, we have put particular emphasis in the introduction to sharing example application areas of SR (25-29).
>
>
>
> > The authors note in the conclusion that future improvements might incorporate “more realistic applications settings” but the paper would be improved if it gave a bit more attention to this point earlier on.
>
> We used some of our additional page to try to clarify what we meant here, lines 320-326:
>
> "In addition, whereas our benchmark includes real-world data as well as simulated data with ground-truth models, it does not include real-world data from phenomena with known, first principles models (e.g., observations of a mass-spring
> damper system). Data such as these could help us better evaluate the ability of SR methods to discover
> relations under real-world conditions. We intend to include these data in future versions, given the
> evidence that SR models can sometimes discover unexpected analytical models that outperform the
> expert models in a field (e.g., in studies of yeast metabolism [76] and fluid tank systems [67])."
>
> Hopefully this clarifies what we meant.
>
>
> > The datasets derived from physics equations are clearly more ‘real’ than contrived toy examples, but are they a good representation of the main ‘real’ context that SR is applied to?
>
> > In particular, the introduction gives little sense of what is at stake in a broader sense in the use of SR methods. I would recommend including some mentions here of the kinds of areas where SR methods have been or could be applied (or perhaps list a few example tasks, more specific than “physics, engineering, statistics…”). In what contexts is it advantageous or necessary to learn an analytical model rather than only predict? Or, do SR algorithms sometimes do better at prediction than ML algorithms and are they being adopted in ML applications? It would also be helpful to have a summary of what the datasets cover. Clearly, the ground-truth datasets come from physics, but what kinds of application areas are the PMLB datasets drawn from? Does it have any relevant gaps, known limitations, or conscious omissions? This can all be fairly brief, but it would help both motivate the paper and make the limitations and social implications of the benchmark more specific.
>
>
> In fact SR methods are applied very broadly to many domains, as we allude to in our referenced applications, and so it is a bit hard to pin down a specific application by area that covers their potential use cases. For that reason, we think applying to a diverse set of problems is warranted. However, a related question is, "under what conditions does one opt to use SR", and for this question we have tried to provide additional motivation. We see SR as a way of searching for interpretable/transparent ML models explicitly (line 62), as has been advocated by Rudin (NMI, 2019). This is in contrast to relying on post-hoc explanations of black-box models, which are limited in their ability to tell the user *why* (by what mechanism) a model makes the decisions it does. If simple enough, SR models are unique in having the potential to be analyzed for their out-of-distribution behavior. In addition, SR models have been used to learn decision-tree like functions in areas like clinical informatics, where the need for transparency is crucial and likely to be regulated by the FDA soon. We mention this motivation in lines 25-29 and 68-69.
>
>
> > There is additional detail on the data in the appendix, but it remains unclear to me (A) whether SR is actively being applied, say, in criminal justice applications (the example dataset context mentioned in the appendix) and (B) whether any information warning of potential biases or contextual factors for particular datasets is provided with the datasets.
>
> At the moment, we do not provide any warning of potential biases for these datasets. It is something we could consider adding to PMLB. As far as we know, SR is not being applied in criminal justice applications, but it is possible.
>
> > This sentence does not make sense to me (pg. 18): “In that regard, we do not foresee it as creating additional ethical issues around their use, unless our work inspires a significant number of people to research symbolic regression.”
>
> Clarified in the revision; the main point being that the crime data (and many other datasets in PMLB) exist in at least two other repositories: e.g. [UCI](https://archive.ics.uci.edu/ml/datasets/communities+and+crime) and [OpenML](https://www.openml.org/d/315).

---

> > ### Comment · Reviewer_m58C · 2021-07-13
> > **Reply**
> >
> > I like the revisions, particularly the increased detail on applications and the added detail in the datasets section. The separate SRBench section is also a helpful change. Happy for this paper to be accepted.

---

### Official Review · Reviewer_DK61 · 2021-07-02
**Benchmarking symbolic regression algorithms**

**Rating:** 6
**Confidence:** 3
**Clarity:** The paper is clearly written and fair…

**Strengths:**

The benchmarking code appears to be easy to apply and the data is made available. The proposed way of benchmarking allows more detailed comparison of symbolic regression methods than other available benchmarking software.

**Weaknesses:**

The concrete advantage over commercial symbolic regression benchmarking platforms (the paper mentions Eureqa and Wolfram) should be stated more clearly. It is also not exactly clear to me how the algorithms used by these platforms differ from the proposed solution and why the proposed solution is more ‘straightforward’ as mentioned in the paper.

Overall, I think the paper spends too little time describing the proposed benchmarking method and its advantages. The summary of existing algorithms which spans a large part of the paper could be shortened to make room for these explanations. The paper’s focus is placed mainly on the benchmarking results and not on the benchmarking methodology. Shifting the focus a little could be beneficial to the wider community given this is a dataset / benchmark track.


**Additional Feedback:**

No

**Correctness:**

To the best of my knowledge, the proposed methodology and experimental setup are correct.

**Documentation:**

The authors provide benchmarking code in a Github repository.

**Relation To Prior Work:**

The authors briefly compare the proposed benchmarking against commercial software for symbolic regression benchmarking, however the differences and advantages could be made clearer.

**Summary And Contributions:**

The authors introduce an open-source benchmarking platform for symbolic regression and benchmark an array of different methods on over 200 regression problems. The benchmarking code and data is readily accessible and appears to be straightforward to use (although I have  not tried). The benchmarking results reveal that the best performing methods for real-world regression combine genetic algorithms with parameter estimation and/or semantic search drivers. In the presence of noise, the authors find that deep learning and genetic algorithm-based approaches perform similarly.

---

> ### Author Response · Authors · 2021-07-12
> **advantages over commercial methods, and benchmark methodology**
>
> Thank you for your comments.
>
> > The concrete advantage over commercial symbolic regression benchmarking platforms (the paper mentions Eureqa and Wolfram) should be stated more clearly. It is also not exactly clear to me how the algorithms used by these platforms differ from the proposed solution and why the proposed solution is more ‘straightforward’ as mentioned in the paper.
>
>  We've edited the text to try to clarify why SRBench is necessary for controlled experiments with SR methods  in lines 77 - 88. It is quite important to note that one cannot truly perform controlled benchmarks when using Eureqa or Wolfram in comparison to a typical open-source SR package, for example, DSR or Operon. Typically in a benchmarking experiment we are very careful to control for the aspects of the algorithm training that could confound the results - for example allowing one algorithm to evaluate 10x as many candidate equations as another. Unfortunately, to my knowledge, this type of fine-grained control simply isn't possible using Eureqa or Wolfram, and so observed differences in benchmarking are hard to pin to any specifics of the algorithm. Furthermore, both are run using cloud compute, which complicates a study design like ours that is able to specify strict CPU and memory limits for each training instance. Finally, both of these commercial tools are closed source, and so even if we were able to benchmark them fairly, we would not necessarily be able to explain differences in performance from a methodological perspective. I hope this now comes across more clearly in the text.
>
> > Overall, I think the paper spends too little time describing the proposed benchmarking method and its advantages. The summary of existing algorithms which spans a large part of the paper could be shortened to make room for these explanations. The paper’s focus is placed mainly on the benchmarking results and not on the benchmarking methodology. Shifting the focus a little could be beneficial to the wider community given this is a dataset / benchmark track.
>
> We agree with this point, and have restructured Section 3 to focus solely on the benchmarking methodology and its advantages. We extended our description of SRBench from 1 to 4 paragraphs. We have also moved the SR method descriptions to Section 4 and shortened them somewhat, but have not significantly shortened them based on the good feedback from reviewer m58C.
>
> Hopefully our revision addresses your concerns. Please reach out with any questions or comments and thank you again.

---

> > ### Comment · Reviewer_DK61 · 2021-07-20
> > **Response to comments**
> >
> > I like the changes and would be happy if this paper is accepted.

---

### Official Review · Reviewer_Vtip · 2021-07-04
**Benchmark for contemporary symbolic regression**

**Rating:** 6
**Confidence:** 3

**Strengths:**

Overall, the paper is clearly-written, with extensive datasets for benchmarking. The authors do a good job at bringing together SR methods and comparing their performance, clearly explaining drawbacks and limitations. The work has good contributions despite the limitations and is a good exposition of presented methods.

**Weaknesses:**

The paper presents limitations in the using real-world data from models, as authors also explain. The presentation of the results in the main text is clear, yet a more in-depth analysis of different datasets and variance across them would be helpful in further understanding the benchmark (although a more extensive analysis is presented in the Appendix). The applications could be strengthened in the exposition as well.

**Additional Feedback:**

None.

**Clarity:**

The paper is very clearly-written, well-structured, and good result visualization.

**Correctness:**

The evaluation methods present limitations but they are explained by the authors.

**Documentation:**

The benchmark presents good documentation through availability of all code used for evaluation and a clear experimental set-up.

**Ethics:**

No.

**Relation To Prior Work:**

The methods used are explained and grounded into previous literature; the set of methods could be expanded in the future, but seems sufficient for the benchmark presented.

**Summary And Contributions:**

The paper presents a benchmark for contemporary symbolic regression, constructing a set of regression problems on which SR methods are tested. The authors use a variety of datasets, some from PMLB, some other SR databases, and use R2 testing for establishing accuracy. Overall, genetic programming-based models seem to perform best, while AIFeynman perform best for finding solutions for synthetic problems.

---

> ### Author Response · Authors · 2021-07-12
> **in-depth analysis of different datasets, variation, and applications**
>
> Thank you for your review, it was very helpful in crafting changes to the manuscript.
>
> > a more in-depth analysis of different datasets and variance across them would be helpful in further understanding the benchmark (although a more extensive analysis is presented in the Appendix).
>
> - Based on your review and the comments of reviewer m58c, we have dedicated extra space to describing the dataset resource in terms of the domains of application, data sources, and bulk dataset properties in lines 225 - 238.
> - By variance across the datasets, we took you to mean variance in performance among methods across the datasets. If you meant another notion of variance, please let us know and we can discuss further.
> - We made two main changes towards describing the variance in performance across datasets with more depth. First, we changed Figure 3 into two panels, each of which shows results for a specific subset of the ground-truth problems. This figure was originally in the Appendix, but we thought it fit well and was also interesting. It shows that, whereas AIFeynman significantly outperforms other SR methods in terms of solution rates for noise levels in [0, 0.001], it does not do so for the Strogatz problems.
> - In addition to changes to Figure 3, we added Figure 7 (p 26), which biclusters the rankings of each method and dataset in the black-box regression benchmark. This figure allows a much more detailed look into where certain methods outperform others, and additionally allows us to analyze the similarity of algorithm behavior on various datasets.  Due to space we couldn't fit this discussion in the main text.
>
> > The applications could be strengthened in the exposition as well.
>
> We have added additional descriptions of the applications of SR in the introduction, lines 24 to 29. We've also described applications to real data for which SR methods found better models than existing expert models (line 326).
>
> Thanks again and please let us know if you have any additional thoughts, questions or comments.

---

> > ### Comment · Reviewer_Vtip · 2021-07-13
> > **Response to comments**
> >
> > Thank you, authors! Agreed that the change to Figure 3 (addition to main text) adds a lot! I appreciate the descriptions of applications that the authors have added (I think an illustrative concrete example might help, but this reads good to me). I'd be happy with having this paper accepted.

---

### Author Response · Authors · 2021-07-12
**Summary of updates in revision 1**

We thank the reviewers for their feedback on the initial manuscript submission which has helped clarify the focus of our paper. Based on the current reviews, we have made the following changes in this version:

1. **Improved descriptions of the datasets (R3)**. We have distinguished and clarified “real-world” versus “synthetic” datasets and summarized the domains, applications and subject areas that are covered.
2. **Additional focus on the SRBench methodology and its advantages (R2)**. Section 3 was heavily expanded to discuss the benchmark methodology.
3. **Motivation for the applications of SR (R1,R3)**. We’ve added additional motivation for why and how SR methods are used.
4. **Advantage of SRBench over commercial SR algorithms (R2)**. We’ve clarified why we think SRBench is crucial for benchmarking SR methods in lieu of using existing commercial solutions.
4. **Variance across datasets (R1)**. We included a breakdown of ground-truth results by data grouping and bi-clustered the variations in performance across all problems and methods. A discussion was added to the appendix due to space.

In addition, we have updated the results for the ground-truth experiments to an additional noise level (powers of ten in the range [0, 1e-3]) and to address a bug in the way that noise was being applied (see [this commit](https://github.com/EpistasisLab/srbench/commit/5667c851d9cc8f2621659b8e9c2e177cb794475c)). The results and conclusions are virtually the same, although updates to the text have been made in accordance with the updated experiment.

---

### Decision · Program_Chairs · 2021-07-26

**Decision:**

Accept

**Comment:**

All reviewers support accepting the paper, especially after taking the author feedback into account. One concern was how the proposed algorithms compare to those implemented in commercial software platforms such as Wolfram. The authors correctly point out that the closed source nature of these platforms makes a comprehensive comparison difficult. A limited comparison could still be a valuable addition to the benchmark so that researchers can compare their algorithms to the commercial state-of-the-art. In any case, the paper is of high quality and I recommend accepting it.